# The Importance of Network Position in the Diffusion of Agricultural Innovations in Smallholders of Dual-Purpose Cattle in Mexico

**Villarroel-Molina Oriana [1]**, **De-Pablos-Heredero Carmen [2]**, **Barba Cecilio [1]**, **Rangel Jaime [3]** and **García Antón [1,\*]**

[1] Animal Science Department, Rabanales University Campus, University of Cordoba, 14071 Cordoba, Spain; z42vimoo@uco.es (V.-M.O.); cjbarba@uco.es (B.C.)

[2] Department of Business Economics (Administration, Management and Organization), Applied Economics II and Fundamentals of Economic Analysis, ESIC Business & Marketing School, Rey Juan Carlos University, Paseo de los Artilleros s/n, 28032 Madrid, Spain; carmen.depablos@urjc.es

[3] Mexico's National Institute for Forestry, Agriculture and Livestock Research (INIFAP), Medellín de Bravo 94277, Mexico; rangel.jaime@inifap.gob.mx

\* Correspondence: pa1gamaa@uco.es

**Abstract:** The dual-purpose bovine production system (DP) is the most widespread small-scale model in Latin American tropics, where it constitutes a key tool in terms of food security. Most DPs are subsistence farms oriented to self-consumption, with a very low technology adoption rate. Hence, the main challenge is how to improve the technological level without compromising the system sustainability by applying land-sharing practices. Thus, through networks methodology, this paper analysed how farmers adopt reproductive technologies. The sample consisted of 383 very small farms of dual-purpose cattle. Seven reproduction technologies oriented to improve reproductive efficiency were evaluated: Breeding soundness evaluation in bulls, semen fertility evaluation, evaluation of female body condition, oestrus detection, pregnancy diagnosis, seasonal or continuous mating, and breeding policy. Social Network Analysis (SNA) allowed identifying adoption patterns, as the joint adoption of semen fertility evaluation, estrus detection, and pregnancy diagnosis, which were consider complementary technologies. Similarly, breeding soundness evaluation in bulls was found to be the most widely adopted technology. The results showed that these farmers presented a very low level of reproduction technology adoption rate and suggested that farmer's affiliation with organizations such as the Livestock Groups for Technological Validation and Transfer (GGAVATT), and its network position had a significant impact on the level of technological adoption. In the first stage of adoption, this work highlighted the importance of centralized models from the GGAVATT to the farmers, related to the knowledge and absorption dynamic capabilities. In a later stage, decentralized models through technological leaders are a priority, related to integration and innovation dynamic capabilities.

**Keywords:** innovations; dual-purpose cattle; technology adoption; centrality network indices; social network analysis

## 1. Introduction

The 2030 Agenda for Sustainable Development is a universal framework for action to end extreme poverty (Goal 1), fight inequality, and address the urgency of climate change and its impacts [1]. In this context, small-scale farms play a crucial role in ensuring food security by widely contributing to the supply and access to food in rural areas [2–4]. Besides, small-scale farms are family-based systems spread worldwide that apply diversified activities (crop and livestock), contributing to improving land use [5]. According to García et al. [6], 85% of small-scale farms are subsistence productions geared primarily to on-farm consumption, while only 15% are market-oriented with commercial objectives.

Therefore, smallholders mainly seek family welfare (including education) and vulnerability reduction by applying low-cost strategies. However, these systems are characterized by low technology adoption levels, low competitiveness, and extreme vulnerability to environmental risks and market changes [7,8]. In developed countries, these issues are compounded by the loss of profitability, lack of generational change, low status in terms of occupational prestige, and the excessive bureaucracy and regulations in rural areas, threatening smallholders' interests and livelihoods [9,10]. Furthermore, in developing countries, thefts and family safety, children's future (employment and education), and the continuous habitat degradation by pesticides and unfriendly agricultural practices are major concerns [2,10,11].

The sustainable development paradigm lies in how to improve productivity in a sustainable way. Understanding how technologies spread among farmers and organizations is required to enable the technological adoption to smallholders. In this scenario, the application of social networks in the diffusion of innovation is a key tool [3,11]. According to Rangel et al. [2] and Mikecz et al. [4], technology adoption was related to size, intensification level, and economic results, pointing out that size is the main factor to determine the technology adoption level. In this regard, dual-purpose (DP) cattle represent an alternative to intensive livestock farming [6,12]. The strengthening of social networks will positively influence the low-cost reproductive technology adoption and its direct application, increasing the system productivity without compromising its sustainability. Faced with a sustainable intensification, with "intensive margin" (marginal cost > mean variable cost), the improvement of productivity with own resources of the system will be possible by friendly practice at minimum cost. In small-scale DP farms, it will be possible develop a low-cost strategy characterized by the low or null opportunity cost of family labor, poor dimension, and "extensive margin" (marginal cost < mean variable cost) [6,7].

The dual-purpose bovine production system (DP) is the most widespread small-scale model in Latin American tropics, in which meat and milk are simultaneously produced. The DP is a genetic cross among native cows (highly adapted to the extreme tropic climate conditions), and European dairy breeds (good milk producers) [2,13,14].

The importance of dual-purpose cattle for sustainable development and land-use efficiency lies in its flexibility in the use of resources, combining agricultural and livestock activities. The DP system uses extensive grazing, with low inputs, low-cost resources such as by-products and crop residues, in a circular economy system process, with land-sharing orientation [15,16]. The land-sharing or wildlife-friendly farming model allows identifying the best technologies and management practices recommended to smallholders [17–19]. In doing so, DP cattle farms contribute to mitigating climate change by reducing greenhouse gas emissions and increasing carbon accumulation [3]. In addition, DP in developing countries has been widely recognized as a pathway out of poverty, a major income-generating activity, and a means of income diversification [6,12,13].

In this context, many solutions to challenges facing global food production and consumption lie in how livestock sectors are managed [20,21]. However, even though the technology is an important tool to increase a sustainable productivity, and despite the efforts made to introduce new technologies into the production system, DP farmers continue to reject technologies that could potentially benefit their livelihood [13,22,23]. The low technology adoption rate has been associated with multiple factors, such as small-dimension, weak financial capacities, lack of technical support, risk aversion, and the misalignment between technological improvements and the farm's objectives, hence the need to understand the reasons for farmer's behaviour towards technology and its low adoption rate.

This research group has previously studied dual-purpose livestock systems in the Mexican tropics. Rangel et al. [15], characterized technologically 1475 farmers, and identified five dual-purpose farmer groups, with high homogeneity within the group and heterogeneity amongst groups. García-Martínez et al. [6] highlighted the need to study the low adoption rate of the most vulnerable group. Furthermore, Villarroel-Molina et al. [3]

showed the usefulness of network analysis to identify key innovations in adoption success and failure factors in livestock systems. Villarroel-Molina et al. [12] used SNA to characterize technological leaders in the genetic area.

This study contributes to deepening the knowledge of the low technology adoption rate in DP livestock systems from a novel perspective, as most of the studies in the field have estimated the network's structure based on adoption intentions [24,25]. As an original approach, this research estimated networks from the technologies already adopted by farmers, identifying DP best practices and technologies.

*Theoretical Background of Social Network Analysis (SNA)*

One of the most central concepts in social network analysis and structural hole theory is the notion of position. On one hand, network position studies refer to the advantages acquired by an actor by occupying a brokering position within a network, so structure is used as an indicator of how information is distributed in a group of people. According to Borgatti and Everett [26], position has played a critical role in the study of the adoption and diffusion of innovation. On the other hand, Reagans et al. [27] and Crandall et al. [28] have pointed out that brokers are potential amplifiers for innovation since they are well positioned to synthesize ideas that arise from different groups of knowledge or technological specialization. Likewise, Burt [29–31] claimed that individuals hold certain positional advantages or disadvantages from how they are embedded in social structures.

The SNA methodology has developed a series of measures that can be included in the processes of agricultural extension to foster innovation. According to Borgatti et al. [32,33], Bonacich [34,35], Opsahl [36], and Freeman [37], the most popular centrality measures are degree, betweenness, eigenvector, and closeness, but we have also included the measure of constraint. Each of these measures quantifies how close each actor is to the central position in the network, but the concept of being central is defined differently in each case [38]. Degree centrality identifies the most popular farmer who shares the maximum number of technologies with other farmers. Closeness centrality identifies the farmers who have the fastest access to information in the network. The top farmers identified by the betweenness centrality are the mediators, brokers, and gatekeepers of communication who can control and influence the diffusion of technologies and innovation in the network. Eigenvector centrality identifies the most influential and authoritative farmers. Constraint is an index that measures the extent to which an actor's contacts are redundant, decreasing the probability of obtaining new information.

Therefore, this research is aimed to address the following research question: How does network position affect the technology adoption of dual-purpose farmers in Mexico?

We believe that obtaining an advantageous position in the network has a positive impact on farmers' technology performance, as it increases the sources of information, allows knowledge-sharing, and facilitates the bringing together of complementary skills from different groups of farmers. This statement is based on social capital theory, which has suggested that people who do better (best practices or higher performances) are somehow better connected [3]. According to the dynamic capabilities approach, technology adoption processes can follow different pathways. Bastanchury-López et al. [17,39], linked this theory with mixed systems performance to understand how best results can be achieved, and found that improvements of dynamic absorption and integration capabilities had a positive impact on performance in dairy sheep farms in Spain. The authors grouped dynamic capabilities into four types: (i) detection capability: the ability to diagnose the environment and understand the needs of the customers better than competitors; (ii) absorption capability: the ability to acknowledge the value of the new, assimilate the information, and apply it to commercial ends; (iii) integration capability: the result of sharing and combining information; and (iv) innovation capability: developing new products and markets through coordination towards a strategic orientation by applying innovative behaviours and processes.

In this context, it is not only the access of agents to information through their network linkages that matters, but their ability to absorb and integrate new technologies. So, the proposals that will be tested are:

- Decentralized approach (peer-to-peer and bottom-up diffusion): A farmer who acts as a broker between two or more closely connected groups of farmers could gain important comparative advantages, performing better than other farmers do. Brokering position allows obtaining new information from other groups, becoming an early adopter in the community. This is the behaviour found by Villarroel-Molina et al. [3] and Zaheer et al. [40].
- Centralized approach (top-down diffusion): In the diffusion of a high cost or complex adoption technology, the organizations act as diffusion agents or facilitators, frequently applying them in pilot farms. In this case, the adoption is homogeneous, and the network structure will be different from the previous one. This is the traditional diffusion approach described in dual-purpose by Espejel-García et al. [41], Espinosa-García et al. [42], and Zarazúa et al. [43].

Both strategies occur simultaneously over time, but it must be considered that the brokering position becomes a comparative advantage only when it happens amongst groups of farmers who provide new or diverse information [3].

Therefore, due to the very low technology adoption rate in DP [3,15], it would be of great interest to go in deep in the knowledge of technology adoption and the diffusion process. Thus, the main purpose of this study was to explore how the reproductive technology adoption process happens and how the smallholder's network position affects the level of technological adoption in very small dual-purpose cattle farms in Mexico. To address this issue, we applied social network analysis (SNA) and examined networks amongst farmers and their implications in reproductive technology dissemination. This study provided insights for practitioners, policymakers, and researchers on actionable strategies and the critical success factors of livestock technology transfer programs in Mexico.

## 2. Materials and Methods

### 2.1. Data Collection

Data were collected from 2013 to 2016 by direct surveys of dual-purpose farmers who have received technical advice from Mexico's Ministry of Agriculture, Livestock, Rural Development, Fisheries and Food (SAGARPA). The sample (*n* = 383) was selected from the 1475 farmers previously described by Rangel et al. [15]; this group represented the more vulnerable small farmers of DP in Mexico, with 50 or fewer cows. The main characteristics of this group of farms have been widely described by Rangel et al. [16] and Villarroel-Molina et al. [3] (Table 1). The annual milk production was 11,229 L, 988 L/cow and 108 L/ha. The farm type had 19.25 Animal Unit (AU) of herd size, 27.17 ha, 1.09 UA/ha of stocking rate. The farmers' mean age was 51 years with three dependent relatives.

**Table 1.** Structural characteristics of dual-purpose farms (*n* = 383).

| Variables | Mean | Median | SD [1] | CV [2] | Min [3] | Max [4] |
|---|---|---|---|---|---|---|
| Grazing surface, ha | 27.17 | 19 | 38.67 | 142.33% | 3 | 400 |
| Total animal unit, AU | 19.25 | 19.2 | 3.96 | 20.57% | 10 | 47 |
| Herd size, n° cattle | 25.54 | 25 | 6.32 | 24.76% | 10 | 65 |
| Stocking rate, UA/ha | 1.09 | 1 | 0.636 | 58.32% | 0.05 | 3.82 |
| Milk production, L/year | 11,229 | 10,000 | 6825 | 60.78% | 0 | 36,500 |
| Milk per cow, L/cow/year | 987.71 | 937.50 | 591.75 | 59.91% | 0 | 2940 |
| Calves sold, n° calves | 4.90 | 4 | 5.81 | 118.56% | 0 | 40 |
| Unproductive animals, heads | 2.53 | 0 | 4.52 | 178.92% | 0 | 32 |
| Cheese yield, kg/farm/year | 245.25 | 0 | 733.71 | 299.17% | 0 | 9000 |
| Milk production, L/ha | 107.80 | 52.63 | 186.79 | 173.27% | 0 | 1429 |
| Stakeholder's age, years | 51 | 51 | 14.51 | 28.40% | 20 | 85 |
| Dependent relatives, n° | 2.91 | 3 | 1.80 | 61.99% | 0 | 9 |
| Employments, workers | 1.49 | 1 | 1.11 | 74.28% | 0 | 6 |

[1] Standard deviation, [2] Coefficient of variation, [3] Minimum, [4] Maximum.

### 2.2. Livestock Innovation Level

We selected the reproductive technologies identified by Rangel et al. [15] and García et al. [6] in very small dual-purpose bovine farms from the Mexican tropics. This area was composed of the following seven technologies oriented to improve reproductive efficiency parameters: Breeding soundness evaluation in bulls (T32), semen fertility evaluation (T33), evaluation of female body condition (T34), estrus detection (T35), pregnancy diagnosis (T36), type of mating (seasonal or continuous) (T37), and breeding policy (T38).

### 2.3. Social Network Analysis Measures

We constructed a two-mode network of 383 farmers and seven reproductive technologies. The farmers were grouped according to the type of organization they were affiliated with, as shown in Figure 1 (two-mode network). This type of network helped us identify technological adoption patterns [3].

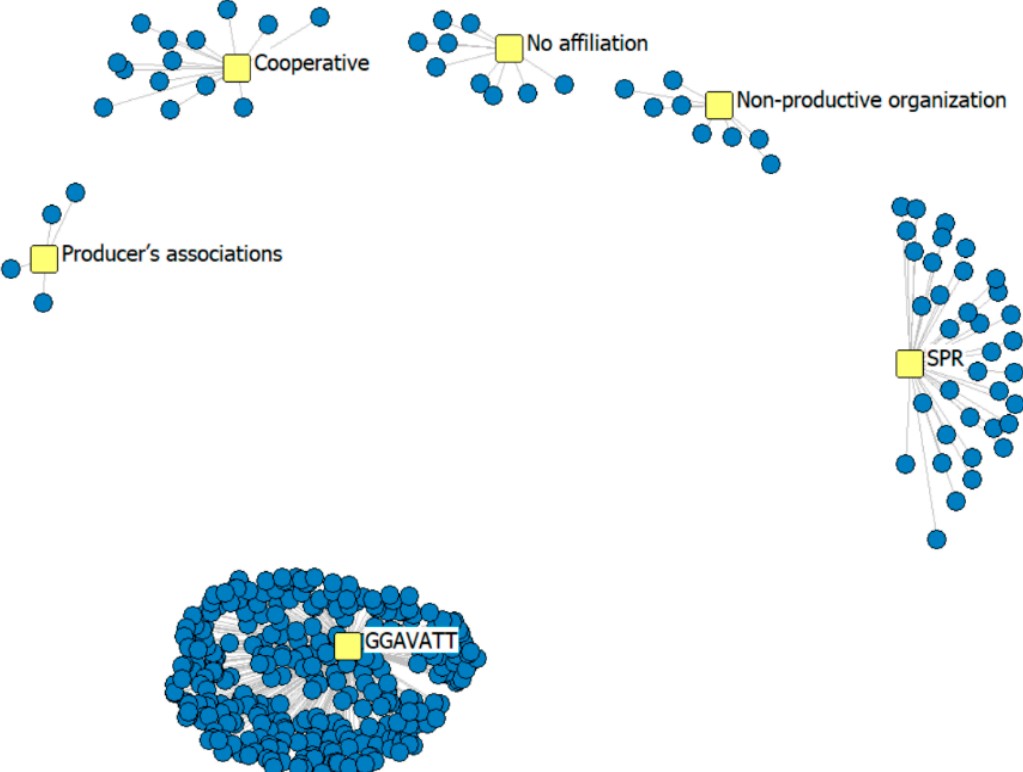

**Figure 1.** Two-mode network visualization of farmers and type of organization. ☐ Organizations: GGAVATT (Livestock Groups for Technological Validation and Transfer), SPR (The Rural Production Society), Cooperative, Producer's organization, Non-productive organization, No affiliation), ● Farmers.

Six different types of organizations were found: The GGAVATT (Livestock Groups for Technological Validation and Transfer) was the most important organization at the public level, belonging to INIFAP, an institution that operates nationally. The GGAVATT constituted the organization with the largest number of affiliated farmers (81.46% of the sample). In the private sector, the most important organization according to the number of affiliates was SPR (The Rural Production Society), with 9.4% of the sample. This group of farmers is also aimed to obtain goods, services, and public or private support to undertake, develop, and consolidate productive and social investment projects. Other less widespread organizations in the area were cooperative society (3.66%), producers' organization (1.04%) and non-productive organization (2.09%). A small group of farmers has no affiliation with any organization (2.35%).

The farmers in the network were highlighted according to the organization type. The initial two-mode network was transformed into a one-mode network through UCINET software, for the network visualization and the identification of the influencer farmers (technological leaders) by considering their centrality network measures. Finally, a comparative benchmarking analysis was carried out among the profiles of five dual-purpose farmers chosen through SNA measures. The analysis and visualization of the dual-purpose cattle network in the Mexican tropics were carried out using UCINET software [44].

## 3. Results

Network measures of descriptive statistics for the sample of 383 farmers are shown in Table 2. The degree obtained was 1.93 related to the low number of technologies that these farmers have adopted, where 50% of smallholders adopted less than one reproductive technology of the seven technologies evaluated. The eigenvector was also very low (0.057), indicating a homogeneous behaviour among farmers and little new information from their closest network. The average betweenness was 8.25, being one of the measures with the highest coefficient of variation (179.42%), related to high heterogeneity in the control of information and degree of influence among farmers. Surprisingly, 50% of the farmers showed a betweenness around 0, while the 25% of farmers presented more than 10.72, the maximum value being 86.76. Network redundant information measured by the level of constraint was very low (0.0117), associated with the network structure itself and the low existing technological level.

**Table 2.** Centrality network measures in dual-purpose farms.

|  | Mean | Standard Error | Median | SD [1] | CV [2] | Q1 | Q3 | Min [3] | Max [4] |
|---|---|---|---|---|---|---|---|---|---|
| Degree | 1.89 | 0.070 | 1 | 1.369 | 72.36 | 1 | 2 | 0 | 7 |
| Closeness | 841.19 | 8.68 | 817 | 169.81 | 20.19 | 806 | 817 | 795 | 1945 |
| Eigenvector | 0.057 | 0.00097 | 0.0462 | 0.019 | 33.43 | 0.046 | 0.065 | 0.014 | 0.108 |
| Betweenness | 8.08 | 0.741 | 0 | 14.49 | 179.42 | 0 | 10.72 | 0 | 86.76 |
| Constraint | 0.012 | 0.0001 | 0.0109 | 0.0028 | 24.20 | 0.010 | 0.011 | 0 | 0.0497 |

[1] Standard deviation; [2] Coefficient of variation, %; [3] Minimum; [4] Maximum.

Technology descriptive statistics are shown in Table 3. An average adoption rate of 27.04% and a coefficient of variation of 109% were found. However, the results showed that breeding soundness evaluation in bulls (96.61%) was the most adopted technology within this area. Seasonal mating was the second most adopted technology, although its adoption rate is considered low (29.50%), and it also presented the highest coefficient of variation (20.85%). Other technologies such as estrus detection (21.41%), pregnancy diagnosis (15.40%), breeding policy (12.27%), and semen fertility evaluation (11.49%) presented very low adoption rates, evaluation of female body condition (2.61%) being the technology with the lowest adoption rate of the group.

**Table 3.** Reproductive technologies in dual-purpose cattle farms.

| Code | Technologies | Mean | SD [1] | CV [2] |
|---|---|---|---|---|
| T32 | Breeding soundness evaluation in bulls | 96.61 | 18.13 | 3.29 |
| T37 | Type of mating | 29.50 | 45.67 | 20.85 |
| T35 | Estrus detection | 21.41 | 41.07 | 16.87 |
| T36 | Pregnancy diagnosis | 15.40 | 36.15 | 13.07 |
| T38 | Breeding policy | 12.27 | 32.85 | 10.79 |
| T33 | Semen fertility evaluation | 11.49 | 31.93 | 10.20 |
| T34 | Evaluation of female body condition | 2.61 | 15.97 | 2.55 |

[1] Standard deviation, [2] Coefficient of variation.

### 3.1. Social Network Analysis Results

Figure 2 shows the dual-purpose farmer's technological innovation network in the reproductive area. To facilitate network visualization and interpretation, only farmers with an adoption rate higher than 40% are shown. The network structure showed that GGAVATT farmers occupy the central positions that are distributed in defined subgroups with other farmers of the same organization (social homogeneity), so there is no evidence of clear leadership. Besides, the structure of the network also showed that farmers from other organizations are distributed peripherally, associated to different groups of farmers (social heterogeneity). Regarding the Producer's associations and Non-productive organization, the farmers f_522 and f_10 stood out, with a betweenness of 19.16 and 16.64, respectively. Furthermore, farmer f_585, who was not affiliated with any organization, was connected to the network with a betweenness of 17.97, his technological strategy was very similar to that of the farmer f_325, belonging to Cooperative, with a betweenness of 17.97.

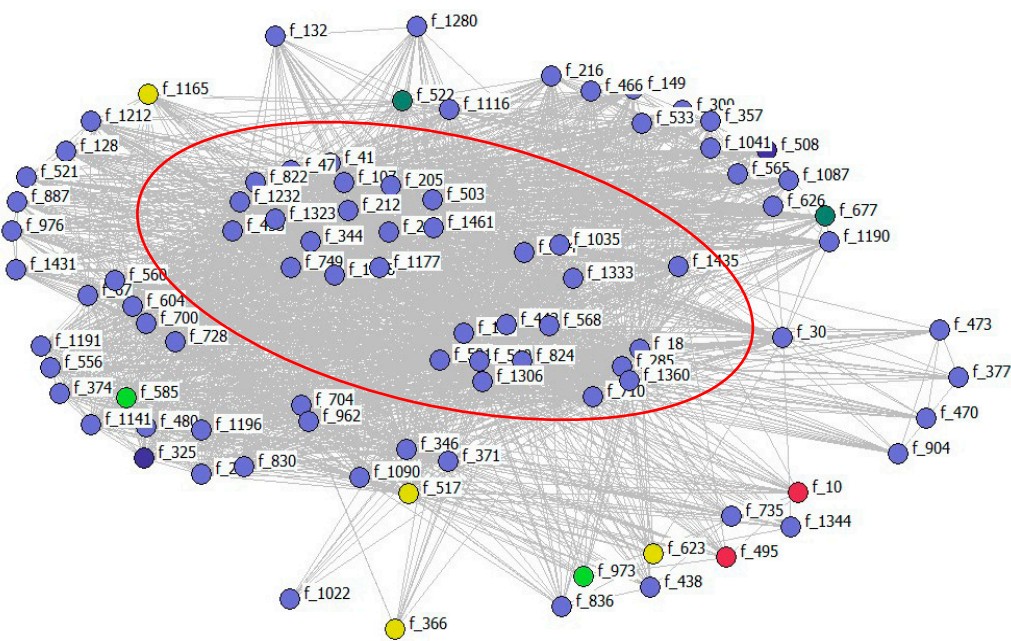

**Figure 2.** One-mode network visualization of farmers with a level of technological adoption greater than 40%. Nodes' colour: organization type. Farmer code (f_$n_i$), organization type: ⬤ GGAVATT, ⬤ SPR, ⬤ Cooperative, ⬤ Producers associations, ⬤ No affiliation, ⬤ Non-productive organization.

To facilitate central farmers and technology leader's identification, only farmers with an adoption rate higher than 70% are shown in Figure 3. In doing so, four leaders with an adoption rate of 100% were identified in the reproductive technological innovation network, all affiliated with the GGAVATT: f_1306, f_824, f_501, and f_510, with the highest betweenness (86.76) and levels of constraints above average (0.0134). The farmers' behavior was analyzed based on the established technological capacity-building, compared to other studies that have evaluated farmers' technological preferences based on adoption intentions [25,45,46]. Besides, f_517 was the only farmer from an organization other than GGAVATT that appears on the network. This farmer was affiliated with SPR, being the leader of this group with an adoption rate of 71.43%, and high betweenness centrality (66.69) much higher than the betweenness of other GGAVATT farmers with lower adoption rates.

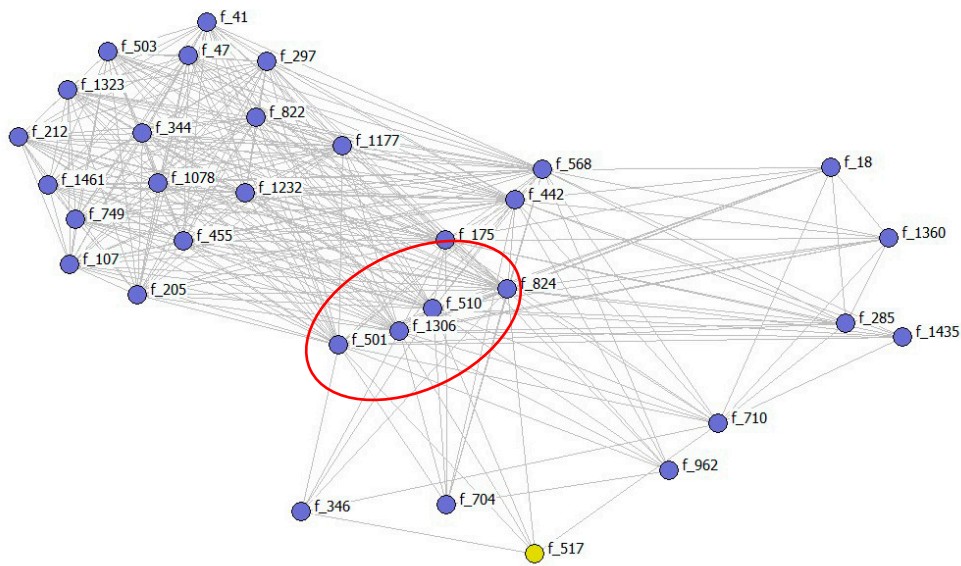

**Figure 3.** One-mode network visualization of farmers with a level of technological adoption greater than 70%. Nodes' colour: organization type. Farmer code (f_n$_i$), organization type: 🔵 GGAVATT, 🟡 SPR, 🟣 Cooperative, 🟢 Producers associations, 🟢 No affiliation, 🔴 Non-productive organization.

Figure 4 showed the two-mode network visualization of farmers and technologies. The network structure indicated that breeding soundness evaluation in bulls (T32) is a basic technology within the reproductive technology package, adopted by most dual-purpose farmers. Another visible adoption pattern is the joint adoption of semen fertility evaluation (T33), estrus detection (T35), and pregnancy diagnosis (T36), for which they are considered complementary technologies.

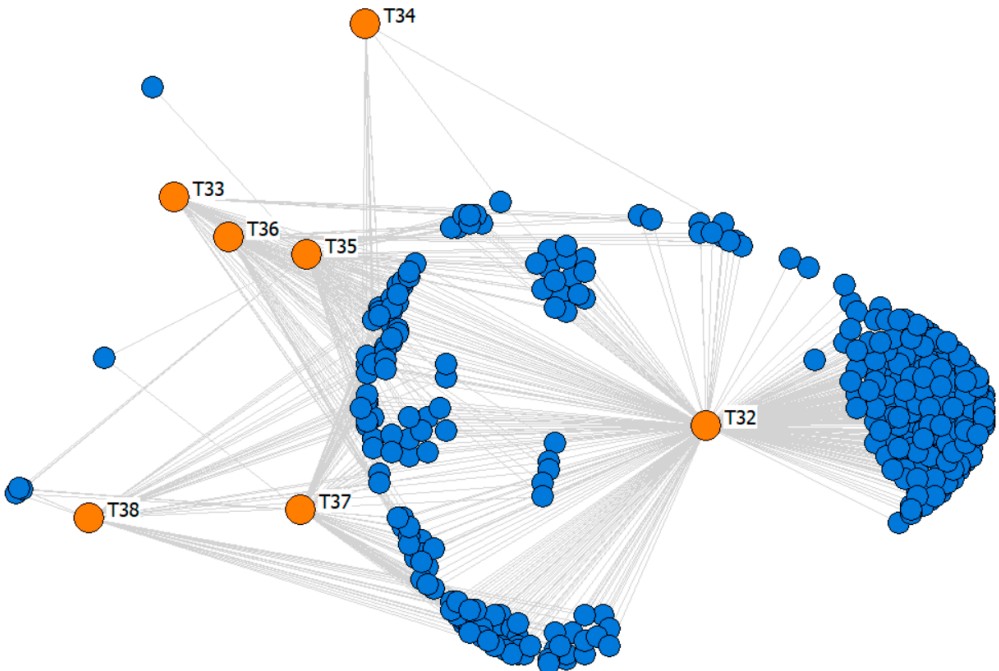

**Figure 4.** Two-mode network visualization of farmers and technologies. 🟠 Technologies. T32, breeding soundness evaluation in bulls. T33, semen fertility evaluation. T34, evaluation of female body condition. T35, oestrus detection. T36, pregnancy diagnosis. T37, type of mating. T38, breeding policy. 🔵 Farmers.

### 3.2. Benchmarking Analysis

The benchmarking of influencer farmers by organization type is shown in Table 4. The leaders were selected according to technological adoption and centrality measures (degree, betweenness, eigenvector, closeness, and constraint). The analysis showed that farmers f_1306 and f_510 were subsistence farmers with more than 70 years and low productive indices, being the first of them meat production-oriented, and the second one milk production-oriented. Farmers f_824 and f_501 are middle-aged producers located in the wet tropic, over 45 years old, who generate two jobs. The first of them was milk production-oriented (1428.57 L/cow/year), and the second showed a mixed productive strategy towards milk and meat. These farmers were affiliated with GGAVATT and showed a high preference for reproductive technologies with an adoption rate of 100% in this technological area. Finally, farmer f_517 affiliated with the SPR was younger (31 years old), showed an adoption rate of 71.43% and a different strategy, combining the sale of cheese and meat.

**Table 4.** Farmer's benchmarking by organization type and technological level.

| Farmer, Code | f_1306 | f_824 | f_501 | f_510 | f_517 |
|---|---|---|---|---|---|
| Organization type | GGAVATT [1] | GGAVATT [1] | GGAVATT [1] | GGAVATT [1] | SPR [2] |
| Technological level, % | 100 | 100 | 100 | 100 | 71.43 |
| Structural characterization | | | | | |
| Ecological zone, tropic | Dry | Wet | Wet | Wet | Dry |
| Productive animals, cows | 12 | 14 | 14 | 10 | 15 |
| Animal unit, heads | 22.8 | 30.8 | 20.5 | 19.20 | 21.9 |
| Stocking rate, UA/ha | 1.333 | 0.993 | 0.891 | 0.96 | 2.701 |
| Grazing surface, ha | 17 | 21 | 20 | 20 | 8 |
| Productive orientation | Meat/subsistence | Milk | Milk/meat | Milk subsistence | Meat/milk |
| Milk production, L/ha | 317.64 | 952.38 | 650 | 605 | 960 |
| Milk yield, L/year | 5400 | 10,000 | 6500 | 6050 | 7680 |
| Milk per cow, L/cow/year | 450 | 1428.57 | 1181.82 | 1210 | 512 |
| Calves sold, n° calves | 5 | 3 | 6 | 4 | 10 |
| Cheese yield, kg/farm/year | 0 | 0 | 0 | 0 | 1000 |
| Stakeholder's age, years | 74 | 57 | 46 | 76 | 31 |
| Dependent relatives, n° | 1 | 2 | 5 | 3 | 3 |
| Employees, workers | 0 | 2 | 2 | 2 | 1 |

[1] Livestock Groups for Technological Validation and Transfer; [2] The Rural Production Society.

## 4. Discussion

Dual-purpose cattle farms were very small, limited to subsistence farming where an important part of the production was oriented to self-consumption, with a highly variable productive strategy (cheese yield, milk, and calves) depending on the immediate environment and the opportunity costs. Given the context in this group of farms, the productive orientation will influence the technology adoption process [7,15,17]. Likewise, affiliation with an organization was considered an important factor in the technology adoption process [3], since organizations are places of access to technological innovations and interaction among farmers where communication flows, constituting a determining factor in the technology capabilities of absorption, integration, and adoption.

The centrality measurements obtained were similar to that of Walther et al. [47], who used SNA to measure the effects of income and gender on informal social networks in the rice value chain, among 490 farmers in Benin, Niger, and Nigeria, and found a degree of 2.5 for women and 3 for men. Our results also agree with the author in eigenvector, both for women (0.03) and for men (0.05). However, regarding betweenness, our results just match the betweenness of 92 found for women, but differ from the 170 found for men. Walter et al. [47] concluded that the monthly profit of farmers was determined by their structural position within the network and the capacity of building connections with other communities outside their ethnic groups and countries. Apart from this, Aguilar-Gallegos

et al. [48] used the SNA approach to study the effects of direct and indirect interactions among 180 rubber producers and key players in the Mexican state of Oaxaca, and found a very low level of betweenness amongst farmers (0.74). This result coincides with 25% of the farmers evaluated in this research, who showed betweenness close to zero. The author relates this behaviour to the fact that some farmers establish links with extension agents rather than with their peers. These results were consistent with structural hole theory and network positioning studies, as high levels of constraints are associated with redundancy in contacts who provide little new information. This circumstance leads farmers to maintain their adoption practices, specializing too much in a specific technological area [11]. The two-mode network makes a significant contribution to advancing the understanding of the reproductive technology low adoption rate as a strategic tool for development and increasing farms competitiveness and viability [6,49]. Based on the adoption patterns identified through SNA, T33, T35, and T36 are complementary technologies. It should be noted that the adoption of one of these technologies will have an impact on the adoption of the other two [3].

The network structure obtained provides a real diagnosis of the most vulnerable farmers' technological situation, and points out the wide technological gap amongst reproduction technologies. In this process, farmers need to know about these technologies to start implementing them on farms (knowledge and absorption capabilities) [17,18]. In this regard, the GGAVATT technicians have made a great effort to enhance the adoption of evaluation in bulls (T32) through technology transfer programs [50]. The potential role of SNA to adopt sustainable innovations is high. Applying this pattern of diffusion could improve the adoption of reproductive technologies. The technologies evaluated were compatible with land-sharing practices, i.e., these are low-cost technologies that do not imply greater intensification or contribution of external inputs and do not present negative impacts on the sustainability of the system [14,18]. In this research SNA and technology were not focused on system intensification but rather towards the improvement of productivity with own farm resources [41,45,49].

In a second stage the implementation of T34, T37, and T38 should be favoured, which requires a productive structure for their adoption [10]. These findings are aligned with those of Aguirre et al. [51], who applied SNA to study the adoption patterns of conservation agriculture practices among 222 maize smallholder farmers in the Mexican state of Chiapas, and found that farmers apply practices that solve emerging problems in the short term, but set aside those practices, which in the medium and long term, would lead to higher and stable yields. However, the network structure differs from that found by Villarroel et al. [3], who evaluated the usefulness of SNA to characterize technology leaders in small dual-purpose cattle farms in Mexico, in the area of genetics, and found a more diverse technological strategy.

The benchmarking analysis showed that farmers affiliated with GGAVATT had competitive advantages in reproduction, such as the access to knowledge (knowledge and absorption capabilities), favouring the technology adoption rate, and a high degree of connectivity with other farmers who belong to the same group [39]. The results showed a centralized pattern of technology adoption (top-down diffusion), where technological adoption, although very low, was concentrated in GGAVATT farmers, with a diffuse network structure and a differentiated leaders' network position. However, these findings differ from Villarroel et al. [3], who found a network structure with well-defined leaders who acted as knowledge disseminators in the organizations and between organizations, identifying decentralized patterns in the diffusion of technology (peer to peer). According to Dhehibi et al. [52] and Zarazúa et al. [24], being connected to technological leaders and farmers affiliated with different organizations (diverse social capital) positively affects the technology adoption rate. Besides, Zarazúa et al. [24] applied SNA to measure social capital amongst corn producers in Mexico and found a strong relationship between the enhancement of technological innovation and the links established by farmers with actors involved in livestock activities. Farmers belonging to the most open network presented bet-

ter productive outcomes and that intermediaries with greater links defined the sources and types of innovation and activated interaction patterns between several actors of the system.

Regarding the theory of dynamic capabilities, Bastanchury et al. [17] and De-Pablos-Heredero et al. [18] studied the stages of the technology adoption process in dairy sheep family farms in Spain, and described the first as a phase of technology knowledge and absorption, the second as an integration stage, and the phase of innovation capability as the final stage. The GGAVATT of a public nature and with the aim of social benefit acts as a knowledge diffuser in the first stage (knowledge and absorption capabilities), with a technology diffusion centralized model (top-down diffusion) and oriented to give technical assistance to the farmers who are affiliated with this organization [43,45]. Due to the farmer's lack of these reproductive technologies, their knowledge and absorption are carried out directly and homogeneously from the GGAVATT technicians to the affiliated farmers through participatory workshops and in situ demonstrations [50]. In the context of innovation dissemination, this is related to the social learning theory, which states that people learn with and from others by example or through observation [53]. Dhehibi et al. [52], pointed out that the know-how influenced the adoption level.

In a second stage of growth and specialization, as described by Villarroel et al. [3] and Aguilar-Gallegos et al. [48], early adopters present a well-defined network structure and are well positioned within the network, standing out as leaders amongst farmers (integration capability). Probably, in a later stage of technological maturity, farmers will start a decentralized diffusion (bottom-down), resulting in innovation processes amongst farmers (peer to peer).

According to the literature and the results, the technology adoption process was sequential and cumulative. In the earlier phase, producer organizations were priorities to technology capabilities of knowledge and absorption. In the technology phase of maturity, capabilities of integration and innovation were provided by the producers, in a decentralized process with highly influential leaders. Thus, the presented findings seem to be consistent with previous research on agricultural technology adoption [3,12,15], which found that farmers' decisions to innovate were based in the context of social interactions among themselves and with agents that promote change.

In summary, SNA can help to understand the actor's behaviour involved in the networks of innovation diffusion, the way these actors engage in the diffusion, and role they play in the adoption of sustainable practices [11,24].

Technological adoption analysis was carried out regardless of farm size, with a minimum cost strategy [2,13,49]. Due to the low level of technology adoption, the system could be improved by favouring the knowledge and absorption capabilities in the less adopted reproductive technologies [6,17], therefore making it possible to increase productivity while maintaining diversified strategies (milk, meat, and cheese) and land-sharing practices in DP farms.

## 5. Conclusions

This research was carried out on the most vulnerable and smallest dual-purpose cattle farmers in the Mexican tropics, who showed a very low level of technology adoption in reproduction. The technology adoption process in the one-mode network was related to factors such as network position, the organization with which farmers were affiliated, and the production strategy. In this research, being affiliated with GGAVATT and connected to technological leaders and farmers was found to positively affect the technology adoption rate. Breeding soundness evaluation in bulls was found to be a widely adopted technology by most farmers (96.61%), while the evaluation of female body condition was the least adopted technology (2.91%). In the two-mode network, another adoption pattern found was the joint adoption of semen fertility evaluation, oestrus detection, and pregnancy diagnosis, for which they were considered complementary technologies, where the adoption of one affects the adoption of the other.

Social network methodology helped to identify key farmers with great potential to spread innovations within their group and amongst other farmers from different groups. SNA also allowed going deep into technology adoption patterns. The social network analysis approach and the dynamic capabilities theory helped to understand the phases of technological adoption. Reproductive technology adoption was in the first technological stage (absorption capabilities), led by GGAVATT in a centralized model (bottom-down). SNA was a useful methodological perspective of analysis to map knowledge networks within smallholder farmer communities that should be undertaken at the planning stage of program development to build community social capital.

It would be of great interest to know the pattern of technological adoption in other areas of dual-purpose cattle farms, such as feeding, pasture management, and milk quality. The main limitation of this methodology lies in knowing the adoption impact of each reproductive technology on the technical performance and the productive and economic outcome. Future studies on the current topic are therefore recommended, using other analyses proposed in the literature to further investigate the network effect on operating results.

**Author Contributions:** Conceptualization and methodology, all authors; formal analysis, software, data curation, data processing, V.-M.O., B.C., and R.J.; statistical and network analysis, V.-M.O., B.C., and G.A.; validation and investigation, G.A. and D.-P.-H.C.; supervision, project administration, G.A., R.J., and D.-P.-H.C.; data acquisition, R.J. and V.-M.O. All authors have been involved in developing, writing, commenting, editing, and reviewing the manuscript. All authors have read and agreed to the published version of the manuscript.

**Funding:** This research received no external funding.

**Institutional Review Board Statement:** Not applicable.

**Informed Consent Statement:** Not applicable.

**Data Availability Statement:** This is not applicable as the data are not in any data repository of public access, however if editorial committee needs access, we will happily provide them, please use this email: pa1gamaa@uco.es.

**Conflicts of Interest:** The authors declare no conflict of interest.

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
