# Peer review of "The Importance of Network Position in the Diffusion of Agricultural Innovations in Smallholders of Dual-Purpose Cattle in Mexico"

_land, doi:10.3390/land10040401_

Round 1
Reviewer 1 Report
This is a very good paper on of network position in the diffusion of agricultural innovations in smallholders of dual-purpose cattle in Mexico. However, several issues should be done for the revision phase:
1) The abstract has to better position this study in the international literature and to highlight what is the paper's added value to existing literature;
2) The theoretical background is quite short and this is in accordance with the reference list (50 reference cited). Authors have to include more studies on broader animal breeding and livestock. For instance, where authors mention that according to the dynamic capabilities approach, technology adoption processes can follow different pathways, with giving example of theory with performance in dairy sheep farms, they can connect also other papers in the filed, highlighting the role of identity for specific farmers as shepherd, cattle growers and their networks etc (eg. Christley et al's paper: https://www.cabdirect.org/cabdirect/abstract/20053202486; see also doi: 10.1080/1070289X.2017.1400322 etc).
3) The methods, results and discussions are nicely presented in the paper, but the conclusions are very broad. There are many results in this study and they could be shortly presented a bit more in relation to what they bring new in current theories. Also, some limitations of the study and follow-up research (e.g. how other studies could follow the results in this paper) is missing in the conclusions.
Author Response
Dear Reviewer,
We want to thank you for your work and valuable comments that have substantially help us to improve the quality of our initial manuscript. We have considered your comments and we have done all the suggested corrections.
Please, find our answer to each of your suggestions in the description below.
We hope you like the new version of the manuscript.
Thank you very much for your attention.
Kind Regards,
The authors
Comments and Suggestions for Authors
- The abstract has to better position this study in the international literature and to highlight what is the paper's added value to existing literature;
Answer: We have added a quick overview about our research to enhance the abstract. We have pointed out the dual-purpose importance in the Latin American tropics and its role in improving food production and increasing sustainability through technological development. So, the new abstract is as follow:
Abstract: The dual-purpose bovine production system (DP) is the most widespread small-scale model in Latin American tropics, where it constitutes a key tool in terms of food security. Most of DP are subsistence farms oriented to self-consumption, with a very low technology adoption rate. Hence, the main challenge is how to improve the technological level without compromising the system sustainability by applying land sharing practices. Thus, through networks methodology, this paper analysed how farmers adopt reproductive technologies. The sample consisted of 383 very small farms of dual-purpose cattle. Seven reproduction technologies oriented to improve reproductive efficiency were evaluated: Breeding soundness evaluation in bulls, semen fertility evaluation, evaluation of female body condition, oestrus detection, pregnancy diagnosis, seasonal o continuous mating, and breeding policy. SNA allowed identifying adoption patterns, as the joint adoption of semen fertility evaluation, oestrus detection, and pregnancy diagnosis, which were consider complementary technologies. Similarly, breeding soundness evaluation in bulls was found to be the most widely adopted technology. The results showed that these farmers presented a very low level of reproduction technologies adoption rate and suggested that farmer's affiliation with organizations such as GAVAATT and its network position had a significant impact on the level of technological adoption. In the first stage of adoption, this work highlighted the importance of centralized models from the GGAVATT to the farmers, related to the knowledge and absorption dynamic capabilities. In a later stage, decentralized models through technological leaders are a priority, related to integration an innovation dynamic capabilities.
2) The theoretical background is quite short and this is in accordance with the reference list (50 reference cited). (eg. Christley et al's paper: https://www.cabdirect.org/cabdirect/abstract/20053202486; see also doi: 10.1080/1070289X.2017.1400322 etc).
Authors have to include more studies on broader animal breeding and livestock. For instance, where authors mention that according to the dynamic capabilities approach, technology adoption processes can follow different pathways, with giving example of theory with performance in dairy sheep farms, they can connect also other papers in the filed, highlighting the role of identity for specific farmers as shepherd, cattle growers and their networks etc
Answer: According to the reviewer suggestions, we have redone the first part of the introduction, pointing out international context challenges, farmer's problems, the small-scale farms importance, the journal interest and how it is approached. Besides, new citations have been added according to his suggestions. The introduction and the theoretical framework have been reordered, and new references to SNA has been included. So, the new introduction is as follow:
- Introduction
The 2030 Agenda for Sustainable Development is a universal framework for action to end extreme poverty (Goal 1), fight inequality, and addressing the urgent of climate change and its impacts [4]. In this context, small-scale farms play a crucial role in ensuring food security by widely contributing to the supply and access to food in rural areas [5-7]. Besides, small-scale farms are family-based systems spread world-wide that apply diversified activities (crop and livestock), contributing to improving land use [8]. According to García et al. [9], 85% of small-scale farms are subsistence productions geared primarily to on-farm consumption, while only 15% are market-oriented with commercial objectives. Therefore, smallholders mainly seek family welfare (including education) and vulnerability reduction by applying low-cost strategies. However, these systems are characterized by low technology adoption levels, low competitiveness, and extreme vulnerability to environmental risks and market changes [10,11]. In developed countries, these issues are compounded by the loss of profitability, lack of generational change, low-status in terms of occupational prestige, and the excessive bureaucracy and regulations in rural areas, threatening smallholders’ interests and livelihoods [12,13]. Furthermore, in developing countries, thefts and family safety, children's future (employment and education), and the continuous habitat degradation by pesticides and unfriendly agricultural practices are major concerns [2,5,13].
The sustainable development paradigm lies in how to improve productivity in a sustainable way. Thus, overcoming this challenge and ensure food security without damaging environmental resources require now more than ever of the promotion of sustainable agricultural and livestock productions that promote efficient land use. In this regard, dual-purpose (DP) cattle represents an alternative to intensive livestock farming [9,14]. The dual-purpose bovine production system (DP) is the most widespread small-scale model in Latin American tropics, in which meat and milk are simultaneously produced. The DP is a genetic cross among native cows (highly adapted to the extreme tropics climate conditions), and European dairy breeds (good milk producers) [5,13,15].
The importance of Dual-purpose cattle for sustainable development and land-use efficiency lies in its flexibility in the use of resources, combining agricultural and livestock activities. DP system uses extensive grazing, with low inputs, low-cost resources such as by-products and crop residues, in a circular economy system process, with land-sharing orientation [16,17]. The land-sharing or wildlife-friendly farming model allows identifying the best technologies and management practices recommended to smallholders [18-20]. In doing so, DP cattle farms contribute to mitigating climate change by reducing greenhouse gas emissions and increasing carbon accumulation [6]. In addition, DP in developing countries has been widely recognized as a pathway out of poverty, a major income-generating activity, and a means of income diversification [9,13,14].
In this context, many solutions to challenges facing global food production and consumption lie in how livestock sectors are managed [3,21]. However, even though, the technology is an important tool to increase a sustainable productivity, and despite the efforts made to introduce new technologies into the production system, DP farmers continue to reject technologies that could potentially benefit their livelihood [22-24]. The low technology adoption rate has been associated with multiple factors, such as small-dimension, weak financial capacities, lack of technical support, risk aversion, and the misalignment between technological improvements and the farm’s objectives. Hence the need to understand the reasons for farmer’s behaviour towards technology and its low adoption rate.
This research group has previously studied dual-purpose livestock systems in the Mexican tropics. Rangel et al. [16], characterized technologically 1475 farmers, and identified five dual-purpose farmers groups, with high homogeneity within the group and heterogeneity amongst groups. García-Martínez et al. [9] highlighted the need to study the low adoption rate of the most vulnerable group. Furthermore, Villarroel-Molina et al. [6] showed the usefulness of networks analysis to identify key innovations adoption success and failure factors in livestock systems. Villarroel-Molina et al. [14] used SNA to characterize technological leaders in the genetic area.
This study contributes to deepening the knowledge of the low technology adoption rate in DP livestock systems from a novel perspective as most of the studies in the field have estimated the network’s structure based on adoption intentions [25,26]. As original approach, this research estimated networks from the technologies already adopted by farmers, identifying DP best practices and technologies.
- Theoretical background of Social Network Analysis (SNA)
One of the most central concepts in social network analysis and structural hole theory is the notion of position. On one hand, networks position studies refer to the advantages acquired by an actor by occupying a brokering position within a network, so structure is used as an indicator of how information is distributed in a group of people. According to Borgatti and Everett [27], position has played a critical role in the study of the adoption and diffusion of innovation. On the other hand, Reagans et al. [28] and Crandall et al. [29], have pointed out that brokers are potential amplifiers for innovation since they are well-positioned to synthesize ideas that arise from different groups of knowledge or technological specialization. Likewise, Burt [30-32], claimed that individuals hold certain positional advantages or disadvantages from how they are embedded in social structures.
The SNA methodology has developed a series of measures that can be included in the processes of agricultural extension to foster innovation. According to Borgatti et al. [33,34], Bonacich [35,36], Opsahl [37], and Freeman [38], the most popular centrality measures are degree, betweenness, eigenvector, and closeness but, we have also included the measure of constraint. Each of these measures quantifies how close each actor is to the central position in the network, but the concept of being central is defined differently in each case [39]. Degree centrality identifies the most popular farmer who shares the maximum number of technologies with other farmers. Closeness centrality identifies the farmers who have the fastest access to information in the network. The top farmers identified by the betweenness centrality are the mediators, brokers, and gate-keepers of communication who can control and influence the diffusion of technologies and innovation in the network. Eigenvector centrality identifies the most influential and authoritative farmers. Constraint is an index that measures the extent to which an ac-tor’s contacts are redundant, decreasing the probability of obtaining new information.
Therefore, this research is aimed to address the following research questions: How does network position affect the technology adoption of dual-purpose farmers in Mexico?
We believe that obtaining an advantageous position on the network has a positive impact on farmers technology performance, as it increases the sources of information, allows knowledge-sharing, and facilitates the bringing together of complementary skills from different groups of farmers. This statement is based on Social capital theory, which has suggested that people who do better (best practices or higher performances) are somehow better connected [6]. According to the dynamic capabilities approach, technology adoption processes can follow different pathways. Bastanchury-López et al. [18,40], linked this theory with mixed systems performance to understand how best results can be achieved, and found that improvements of dynamic absorption and integration capabilities had a positive impact on performance in dairy sheep farms in Spain. The authors grouped dynamic capabilities into four types: (i) detection capability: the ability to diagnose the environment and understand the needs of the customers better than competitors; (ii) absorption capability: the ability to acknowledge the value of the new, assimilate the information, and apply it to commercial ends; (iii) integration capability: it is the result of sharing and combining information; (iv) innovation capability: developing new products and markets, through coordination towards a strategic orientation by applying innovative behaviours and processes.
In this context, it is not only the access of agents to information through their network linkages what matters, but their ability to absorb and integrate new technologies. So, the proposals that will be tested are:
- Decentralized approach (Peer-to-peer and bottom-up diffusion): A farmer who acts as a broker between two or more closely connected groups of farmers could gain important comparative advantages, performing better than other farmers do. Brokering position allows obtaining new information from other groups, becoming an early adopter in the community. This is the behaviour found by Villarroel-Molina et al. [6], and Zaheer et al. [41].
- Centralized approach (top-down diffusion): In the diffusion of a high cost or complex adoption technology, the organizations act as diffusion agents or facilitators, frequently applying them in pilot farms. In this case, the adoption is homogeneous, and the network structure will be different from the previous one.This is the traditional diffusion approach described in dual-purpose by Espejel-García et al. [42], Espinosa-García et al. [43], and Zarazúa et al. [44].
Both strategies occur simultaneously over time, but it must be considered that brokering position becomes a comparative advantage only when it happens amongst groups of farmers who provide new or diverse information. [6].
Therefore, due to the very low technology adoption rate in DP [6,24], it would be of great interest to go in deep in the knowledge of technologies adoption and diffusion process. Thus, the main purpose of this study was to explore how the reproductive technology adoption process happens and how the smallholder’s network position affects the level of technological adoption in very small dual-purpose cattle farms in Mexico. To address this issue, we applied social network analysis (SNA) and examined networks amongst farmers and their implications in reproductive technologies dissemination. This study provided insights for practitioners, policymakers, and researchers on actionable strategies and the critical success factors of livestock technology transfer programs in Mexico.
“1The methods, results and discussions are nicely presented in the paper, but the conclusions are very broad. There are many results in this study and they could be shortly presented a bit more in relation to what they bring new in current theories. Also, some limitations of the study and follow-up research (e.g. how other studies could follow the results in this paper) is missing in the conclusions.
Answer: According to the reviewer's suggestion: The results and discussion have been split into two parts, and we have broadened the SNA relations to dynamic capabilities theory. Besides the technology adoption process, and the farmers and producer organizations role have been described in greater depth. The conclusions have been rearranged and expanded to other issues of greater practical interest, highlighting their novel aspects. Some limitations of the study and follow-up research have also been pointed out. So, the new conclusions are shown below:
According to the literature and the results, the technology adoption process was sequential and cumulative. In the earlier phase, producer organizations were priorities to technology capabilities of knowledge and absorption. In the technology phase of maturity, capabilities of integration and innovation were provided by the producers, in a decentralized process with highly influential leaders. Thus, the presented findings seem to be consistent with previous research on agricultural technology adoption [6,14,16], which found that farmers’ decisions to innovate were based in the context of social interactions among themselves and with agents that promote change.
In summary, technological adoption analysis was carried out regardless of farms size, with a minimum cost strategy [5,22,51]. Due to the low level of technology adoption, the system could be improved by favouring the knowledge and absorption capabilities in the less adopted reproductive technologies [9,18]. Therefore, making it possible to increase productivity while maintaining diversified strategies (milk, meat and cheese) and land-sharing practices in DP farms.
- Conclusions
This research was carried out on the most vulnerable and smallest dual-purpose cattle farmers in the Mexican tropics, who showed a very low level of technology adoption in reproduction. The technology adoption process in one-mode network, was related to factors such as network position, the organization in which farmers were affiliated to, and the production strategy. In this research, being affiliated to GGAVATT and connected to technological leaders and farmers was found to positively affect the technology adoption rate. Breeding soundness evaluation in bulls was found to be a widely adopted technology by most farmers (96.61%), while the evaluation of female body condition was the least adopted technology (2.91%). In two-mode network, another adoption pattern found was the joint adoption of semen fertility evaluation, oestrus detection, and pregnancy diagnosis, for which they were considered complementary technologies, where the adoption of one affects the adoption of the other.
Social networks methodology helped to identify key farmers with great potential to spread innovations within their group and amongst other farmers from different groups. SNA also allowed going into deep in technology adoption patterns. The social network analysis approach and the the dynamic capabilities’ theory helped to understand the phases of technological adoption. Reproductive technologies adoption was in the first technological stage (absorption capabilities), led by GGAVATT in a centralized model (button-down). SNA was a useful methodological perspective of analysis to map knowledge networks within smallholder farmer communities that should be under-taken at the planning stage of programme development to build community social capital.
It would be of great interest to know the pattern of technological adoption in other areas of dual-purpose cattle farms, such as feeding, pasture management and milk quality. The main limitation of this methodology lies in knowing the adoption impact of each reproductive technology on the technical performance and the productive and economic outcome. Future studies on the current topic are therefore recommended, using other analyses proposed in the literature to further investigate the network effect on operating results.

Reviewer 2 Report
This manuscript examines a series of seven technologies for improvement of reproductive efficiency in cattle, within a dataset for 383 small farms in Mexico. The analysis uses Social Network Analysis to investigate the data.
Abstract.
Perhaps start with what you studied (networks of interaction and relationships as factors influencing uptake of technologies to improve cattle reproductive efficiency in small farms).
line 18. Define SNA in full (Social Network Analysis) the first time it is mentioned in the manuscript, the abbreviation thereafter.
The abstract should be revised to address the relevance of the research and results found to land use (within small cattle farms in Mexico). The paper is submitted to LAND - the authors should be aware of the interests of the readers of LAND, which, for many, is not reproductive technologies in cattle. Material relating cattle production to land use is contained in the Introduction; a precis of this could usefully be included in the Abstract. Essentially, a reader who only reads the Abstract should understand how the subject and results of this paper relate to land use.
line 34. Meat (not meet).
Lines 34-6. Is there a citation to support the assertion that milk feeds the population (why not men?)
line 37. making Mexico (not being Mexico)
line 49. What is "Its" in this paragraph? Presumably it is Dual Purpose cattle.
Lines 60-63. These describe the motivation and purpose of the study. Consider placing this earlier in the Introduction (and also the Abstract - see above).
Lines 64-71 describe your previous research and Lines 72-80 the research here.
Lines 81-120 outline relevant theory and context. The impact of this paper would be improved by moving this material to earlier in the Intorduction, since this understanding of network position is central to a wider range of technological adaptation and innovation than the example presented in the paper. Readers of LAND would benefit from this context and insight, over and above the detail of the case study in the paper.
Line 135. 'direct surveys of dual-purpose' (not 'direct surveys done to dual-purpose)
line 138. ; instead of ,
Figure 1. What defines position in this figure? What defines the x,y plane and the specific position of each dot? Can it include axes and scales? This would be helpful, otherwise it is just groups of dots without any context that links them and their aggregation/separation.
line 237-9. Presumably this comment is about hte study by Aguilar-Gallegos et al? How does it relate to your results? Is the network density low in your case study, leading to a similar result? Or does the similarity of result lead you to conclude the same as Aguilar-Gallegos et al?
line 243. Seasonal or continuous? Also in Table 2 (T37)
line 247. with (not 'being')
Figures 2 and 3.
a) as with Figure 1, what defines the position of dots and scaling?
b) consider changing the colours to a scheme that is more widely (and easily) read. The blue (for GGAVATT) and purple (for Cooperative) are very close. Similarly, the red (Producers) and orange (no-affiliation) are likely to be confused with each other, and also with the green (non-producers) under various forms of colour-blindness. Perhaps consider VIRIDIS or similar colour scales?
Results and Discussion
The presentation of the results and discussion of meaning, especially with the readers in mind, would be improved by splitting the material into separate Results and Discussion sections. This would allow the authors to describe their results and then explore the lessons learned for adoption of technologies within networks within the Discussion. I think that such a revision would improve the impact of the work, allowing the wider relevance of the detailed work carried out to be explored for readers of LAND.
Overall, this is an interesting paper, although the authors should be encouraged to revise the manuscript with the readers of LAND more directly in mind. I hope that i have indicated ways to increase the relevance to readers of LAND in my comments above. If the paper can be revised to highlight the lessons of social network analysis and make some links to the wider role of this in land use for food production, then this paper is suitable for LAND.
One further observation. The introduction talks about the need to increase milk production in Mexico (a major importer of dairy in Latin America - line 37) but the case study is of the very smallest and most vulnerable cattle farms (lines 138-9). What is the role of these farms in increasing domestic production of dairy in Mexico? The role of small DP farms in Mexico should be explored a little in the Discussion
Author Response
Dear Reviewer,
We want to thank you for your work and valuable comments that have substantially help us to improve the quality of our initial manuscript. We have considered your comments and we have done all the suggested corrections.
Please, find our answer to each of your suggestions in the description below.
We hope you like the new version of the manuscript.
Thank you very much for your attention.
Kind Regards,
The authors
Comments and Suggestions for Authors
This manuscript examines a series of seven technologies for improvement of reproductive efficiency in cattle, within a dataset for 383 small farms in Mexico. The analysis uses Social Network Analysis to investigate the data.
Abstract.
Perhaps start with what you studied (networks of interaction and relationships as factors influencing uptake of technologies to improve cattle reproductive efficiency in small farms). The abstract should be revised to address the relevance of the research and results found to land use (within small cattle farms in Mexico). The paper is submitted to LAND - the authors should be aware of the interests of the readers of LAND, which, for many, is not reproductive technologies in cattle. Material relating cattle production to land use is contained in the Introduction; a precis of this could usefully be included in the Abstract. Essentially, a reader who only reads the Abstract should understand how the subject and results of this paper relate to land use.
Answer: We have added a quick overview about our research to enhance the abstract. We have pointed out the dual-purpose importance in the Latin American tropics and its role in improving food production and increasing sustainability through technological development. So, the new abstract is as follow:
Abstract: The dual-purpose bovine production system (DP) is the most widespread small-scale model in Latin American tropics, where it constitutes a key tool in terms of food security. Most of DP are subsistence farms oriented to self-consumption, with a very low technology adoption rate. Hence, the main challenge is how to improve the technological level without compromising the system sustainability by applying land sharing practices. Thus, through networks methodology, this paper analysed how farmers adopt reproductive technologies. The sample consisted of 383 very small farms of dual-purpose cattle. Seven reproduction technologies oriented to improve reproductive efficiency were evaluated: Breeding soundness evaluation in bulls, semen fertility evaluation, evaluation of female body condition, oestrus detection, pregnancy diagnosis, seasonal o continuous mating, and breeding policy. SNA allowed identifying adoption patterns, as the joint adoption of semen fertility evaluation, oestrus detection, and pregnancy diagnosis, which were consider complementary technologies. Similarly, breeding soundness evaluation in bulls was found to be the most widely adopted technology. The results showed that these farmers presented a very low level of reproduction technologies adoption rate and suggested that farmer's affiliation with organizations such as GAVAATT and its network position had a significant impact on the level of technological adoption. In the first stage of adoption, this work highlighted the importance of centralized models from the GGAVATT to the farmers, related to the knowledge and absorption dynamic capabilities. In a later stage, decentralized models through technological leaders are a priority, related to integration an innovation dynamic capabilities.
line 18. Define SNA in full (Social Network Analysis) the first time it is mentioned in the manuscript, the abbreviation thereafter.
Answer: We have done the suggested modification. The first time SNA is mentioned at the end of the introduction this way:
Villarroel-Molina et al. [6] showed the usefulness of Social Network Analysis (SNA) to identify key innovations adoption success and failure factors in livestock systems. Villarroel-Molina et al. [14] used SNA to characterize technological leaders in the genetic area.
line 34. Meat (not meet).
Answer: We have done the suggested change, and now the new sentence is as follow:
The future is uncertain and dietary changes, such as eating more protein and meat, are driving up animal-source food request.
line 37. making Mexico (not being Mexico)
Answer: We have included the change and now the new sentence is as follow:
However, domestic milk production and milk products are insufficient, making it necessary to import around 37% of national consumption [2], making Mexico one of the major dairy product importers in Latin America.
line 49. What is "Its" in this paragraph? Presumably it is Dual Purpose cattle.
Answer: We have changed its by “The importance of Dual-purpose cattle for sustainable development and land-use efficiency lies in its flexibility in the use of resources, combining agricultural and livestock activities.”
The following comments are all related to the introduction. So, they will be answered together to make it more readable and comprehensible.
Lines 34-6. Is there a citation to support the assertion that milk feeds the population (why not men?)
Lines 60-63. These describe the motivation and purpose of the study. Consider placing this earlier in the Introduction (and also the Abstract - see above).
Lines 64-71 describe your previous research and Lines 72-80 the research here.
Lines 81-120 outline relevant theory and context. The impact of this paper would be improved by moving this material to earlier in the Introduction, since this understanding of network position is central to a wider range of technological adaptation and innovation than the example presented in the paper. Readers of LAND would benefit from this context and insight, over and above the detail of the case study in the paper.
Answer: According to the reviewer suggestions, we have redone the first part of the introduction, pointing out international context challenges, farmer's problems, the small-scale farms importance, the journal interest and how it is approached. Besides, new citations have been added according to the suggestions. The introduction and the theoretical framework have been reordered, and new references to SNA have been included. So, the new introduction is as follow:
- Introduction
The 2030 Agenda for Sustainable Development is a universal framework for action to end extreme poverty (Goal 1), fight inequality, and addressing the urgent of climate change and its impacts [4]. In this context, small-scale farms play a crucial role in ensuring food security by widely contributing to the supply and access to food in rural areas [5-7]. Besides, small-scale farms are family-based systems spread world-wide that apply diversified activities (crop and livestock), contributing to improving land use [8]. According to García et al. [9], 85% of small-scale farms are subsistence productions geared primarily to on-farm consumption, while only 15% are market-oriented with commercial objectives. Therefore, smallholders mainly seek family welfare (including education) and vulnerability reduction by applying low-cost strategies. However, these systems are characterized by low technology adoption levels, low competitiveness, and extreme vulnerability to environmental risks and market changes [10,11]. In developed countries, these issues are compounded by the loss of profitability, lack of generational change, low-status in terms of occupational prestige, and the excessive bureaucracy and regulations in rural areas, threatening smallholders’ interests and livelihoods [12,13]. Furthermore, in developing countries, thefts and family safety, children's future (employment and education), and the continuous habitat degradation by pesticides and unfriendly agricultural practices are major concerns [2,5,13].
The sustainable development paradigm lies in how to improve productivity in a sustainable way. Thus, overcoming this challenge and ensure food security without damaging environmental resources require now more than ever of the promotion of sustainable agricultural and livestock productions that promote efficient land use. In this regard, dual-purpose (DP) cattle represents an alternative to intensive livestock farming [9,14]. The dual-purpose bovine production system (DP) is the most widespread small-scale model in Latin American tropics, in which meat and milk are simultaneously produced. The DP is a genetic cross among native cows (highly adapted to the extreme tropics climate conditions), and European dairy breeds (good milk producers) [5,13,15].
The importance of Dual-purpose cattle for sustainable development and land-use efficiency lies in its flexibility in the use of resources, combining agricultural and livestock activities. DP system uses extensive grazing, with low inputs, low-cost resources such as by-products and crop residues, in a circular economy system process, with land-sharing orientation [16,17]. The land-sharing or wildlife-friendly farming model allows identifying the best technologies and management practices recommended to smallholders [18-20]. In doing so, DP cattle farms contribute to mitigating climate change by reducing greenhouse gas emissions and increasing carbon accumulation [6]. In addition, DP in developing countries has been widely recognized as a pathway out of poverty, a major income-generating activity, and a means of income diversification [9,13,14].
In this context, many solutions to challenges facing global food production and consumption lie in how livestock sectors are managed [3,21]. However, even though, the technology is an important tool to increase a sustainable productivity, and despite the efforts made to introduce new technologies into the production system, DP farmers continue to reject technologies that could potentially benefit their livelihood [22-24]. The low technology adoption rate has been associated with multiple factors, such as small-dimension, weak financial capacities, lack of technical support, risk aversion, and the misalignment between technological improvements and the farm’s objectives. Hence the need to understand the reasons for farmer’s behaviour towards technology and its low adoption rate.
This research group has previously studied dual-purpose livestock systems in the Mexican tropics. Rangel et al. [16], characterized technologically 1475 farmers, and identified five dual-purpose farmers groups, with high homogeneity within the group and heterogeneity amongst groups. García-Martínez et al. [9] highlighted the need to study the low adoption rate of the most vulnerable group. Furthermore, Villarroel-Molina et al. [6] showed the usefulness of networks analysis to identify key innovations adoption success and failure factors in livestock systems. Villarroel-Molina et al. [14] used SNA to characterize technological leaders in the genetic area.
This study contributes to deepening the knowledge of the low technology adoption rate in DP livestock systems from a novel perspective as most of the studies in the field have estimated the network’s structure based on adoption intentions [25,26]. As original approach, this research estimated networks from the technologies already adopted by farmers, identifying DP best practices and technologies.
- Theoretical background of Social Network Analysis (SNA)
One of the most central concepts in social network analysis and structural hole theory is the notion of position. On one hand, networks position studies refer to the advantages acquired by an actor by occupying a brokering position within a network, so structure is used as an indicator of how information is distributed in a group of people. According to Borgatti and Everett [27], position has played a critical role in the study of the adoption and diffusion of innovation. On the other hand, Reagans et al. [28] and Crandall et al. [29], have pointed out that brokers are potential amplifiers for innovation since they are well-positioned to synthesize ideas that arise from different groups of knowledge or technological specialization. Likewise, Burt [30-32], claimed that individuals hold certain positional advantages or disadvantages from how they are embedded in social structures.
The SNA methodology has developed a series of measures that can be included in the processes of agricultural extension to foster innovation. According to Borgatti et al. [33,34], Bonacich [35,36], Opsahl [37], and Freeman [38], the most popular centrality measures are degree, betweenness, eigenvector, and closeness but, we have also included the measure of constraint. Each of these measures quantifies how close each actor is to the central position in the network, but the concept of being central is defined differently in each case [39]. Degree centrality identifies the most popular farmer who shares the maximum number of technologies with other farmers. Closeness centrality identifies the farmers who have the fastest access to information in the network. The top farmers identified by the betweenness centrality are the mediators, brokers, and gate-keepers of communication who can control and influence the diffusion of technologies and innovation in the network. Eigenvector centrality identifies the most influential and authoritative farmers. Constraint is an index that measures the extent to which an ac-tor’s contacts are redundant, decreasing the probability of obtaining new information.
Therefore, this research is aimed to address the following research questions: How does network position affect the technology adoption of dual-purpose farmers in Mexico?
We believe that obtaining an advantageous position on the network has a positive impact on farmers technology performance, as it increases the sources of information, allows knowledge-sharing, and facilitates the bringing together of complementary skills from different groups of farmers. This statement is based on Social capital theory, which has suggested that people who do better (best practices or higher performances) are somehow better connected [6]. According to the dynamic capabilities approach, technology adoption processes can follow different pathways. Bastanchury-López et al. [18,40], linked this theory with mixed systems performance to understand how best results can be achieved, and found that improvements of dynamic absorption and integration capabilities had a positive impact on performance in dairy sheep farms in Spain. The authors grouped dynamic capabilities into four types: (i) detection capability: the ability to diagnose the environment and understand the needs of the customers better than competitors; (ii) absorption capability: the ability to acknowledge the value of the new, assimilate the information, and apply it to commercial ends; (iii) integration capability: it is the result of sharing and combining information; (iv) innovation capability: developing new products and markets, through coordination towards a strategic orientation by applying innovative behaviours and processes.
In this context, it is not only the access of agents to information through their network linkages what matters, but their ability to absorb and integrate new technologies. So, the proposals that will be tested are:
- Decentralized approach (Peer-to-peer and bottom-up diffusion): A farmer who acts as a broker between two or more closely connected groups of farmers could gain important comparative advantages, performing better than other farmers do. Brokering position allows obtaining new information from other groups, becoming an early adopter in the community. This is the behaviour found by Villarroel-Molina et al. [6], and Zaheer et al. [41].
- Centralized approach (top-down diffusion): In the diffusion of a high cost or complex adoption technology, the organizations act as diffusion agents or facilitators, frequently applying them in pilot farms. In this case, the adoption is homogeneous, and the network structure will be different from the previous one.This is the traditional diffusion approach described in dual-purpose by Espejel-García et al. [42], Espinosa-García et al. [43], and Zarazúa et al. [44].
Both strategies occur simultaneously over time, but it must be considered that brokering position becomes a comparative advantage only when it happens amongst groups of farmers who provide new or diverse information. [6].
Therefore, due to the very low technology adoption rate in DP [6,24], it would be of great interest to go in deep in the knowledge of technologies adoption and diffusion process. Thus, the main purpose of this study was to explore how the reproductive technology adoption process happens and how the smallholder’s network position affects the level of technological adoption in very small dual-purpose cattle farms in Mexico. To address this issue, we applied social network analysis (SNA) and examined networks amongst farmers and their implications in reproductive technologies dissemination. This study provided insights for practitioners, policymakers, and researchers on actionable strategies and the critical success factors of livestock technology transfer programs in Mexico.
Line 135. 'direct surveys of dual-purpose' (not 'direct surveys done to dual-purpose)
Answer: We have done the suggested change, and now the new sentence is as follow:
Data were collected from 2013 to 2016 by direct surveys of dual-purpose farmers who have received technical advice from Mexico’s Ministry of Agriculture, Livestock, Rural Development, Fisheries and Food (SAGARPA).
line 138. ; instead of ,
We have replaced (,) by (;) where it were required.
Figure 1. What defines position in this figure? What defines the x,y plane and the specific position of each dot? Can it include axes and scales? This would be helpful, otherwise it is just groups of dots without any context that links them and their aggregation/separation.
Answer: Unfortunately, UCINET program does not allow including axes and scales in figures. In this type of analysis, the most important thing is how close the nodes are positioned to each other, as it refers to the similarity or dissimilarity in behavior.
line 237-9. Presumably this comment is about the study by Aguilar-Gallegos et al? How does it relate to your results? Is the network density low in your case study, leading to a similar result? Or does the similarity of result lead you to conclude the same as Aguilar-Gallegos et al?
line 243. Seasonal or continuous? Also in Table 2 (T37)
It has been corrected in text: in method: type of mating (seasonal or continuous).
line 247. with (not 'being')
Answer: We have replaced (being) by (with) where it were required.
Figures 2 and 3.
- a) as with Figure 1, what defines the position of dots and scaling?
- b) consider changing the colours to a scheme that is more widely (and easily) read. The blue (for GGAVATT) and purple (for Cooperative) are very close. Similarly, the red (Producers) and orange (no-affiliation) are likely to be confused with each other, and also with the green (non-producers) under various forms of colour-blindness. Perhaps consider VIRIDIS or similar colour scales?
Answer: As we have pointed out previously, the position is defined by similarities or dissimilarities in technology adoption behavior. We have changed the color of the nodes as recommended. The new figures are as follow:
Results and Discussion
The presentation of the results and discussion of meaning, especially with the readers in mind, would be improved by splitting the material into separate Results and Discussion sections. This would allow the authors to describe their results and then explore the lessons learned for adoption of technologies within networks within the Discussion. I think that such a revision would improve the impact of the work, allowing the wider relevance of the detailed work carried out to be explored for readers of LAND.
Overall, this is an interesting paper, although the authors should be encouraged to revise the manuscript with the readers of LAND more directly in mind. I hope that i have indicated ways to increase the relevance to readers of LAND in my comments above. If the paper can be revised to highlight the lessons of social network analysis and make some links to the wider role of this in land use for food production, then this paper is suitable for LAND.
One further observation. The introduction talks about the need to increase milk production in Mexico (a major importer of dairy in Latin America - line 37) but the case study is of the very smallest and most vulnerable cattle farms (lines 138-9). What is the role of these farms in increasing domestic production of dairy in Mexico? The role of small DP farms in Mexico should be explored a little in the Discussion.
Answer: According to the reviewer’s suggestion: The results and discussion have been split into two parts, and we have broadened the SNA relations to dynamic capabilities theory. Besides, the technology adoption process, and the farmers and producer organizations role has been described in greater depth. The conclusions have been rearranged and expanded to other issues of greater practical interest, highlighting novel aspects. Some limitations of the study and follow-up research have also been pointed out. So, the new conclusions are shown below:
- Results
Network measures of descriptive statistics for the sample of 383 farmers are shown in table 2. The degree obtained was 1.93 related to the low number of technologies that these farmers have adopted, where 50% of smallholders adopted less than one reproductive technology of the seven technologies evaluated. The eigenvector was also very low (0.057), indicating a homogeneous behaviour among farmers and little new information from their closest network. The average betweenness was 8.25, being one of the measures with the highest coefficient of variation (179.42%), related to high heterogeneity in the control of information and degree of influence among farmers. Surprisingly, 50% of the farmers showed a betweenness around 0, while the 25% of farmers presented more than 10.72, being the maximum value of 86.76. Network redundant information measured by the level of constraint was very low (0.0117), associated with the network structure itself and the low existing technological level.
Table 1. Centrality network measures in dual-purpose farms.
|
Mean |
Standard error |
Median |
SD1 |
CV2 |
Q1 |
Q3 |
Min3 |
Max4 |
|
|
Degree |
1.89 |
0.070 |
1 |
1.369 |
72.36 |
1 |
2 |
0 |
7 |
|
Closeness |
841.19 |
8.68 |
817 |
169.81 |
20.19 |
806 |
817 |
795 |
1945 |
|
Eigenvector |
0.057 |
0.00097 |
0.0462 |
0.019 |
33.43 |
0.046 |
0.065 |
0.014 |
0.108 |
|
Betweenness |
8.08 |
0.741 |
0 |
14.49 |
179.42 |
0 |
10.72 |
0 |
86.76 |
|
Constraint |
0.012 |
0.0001 |
0.0109 |
0.0028 |
24.20 |
0.010 |
0.011 |
0 |
0.0497 |
1Standard deviation, 2Coecient of variation, %, 3Minimum, 4Maximum.
Technology descriptive statistics are shown in Table 3. An average adoption rate of 27.04%, and a coefficient of variation of 109% were found. However, the results showed that breeding soundness evaluation in bulls (96.61%) was the most adopted technology within this area. Seasonal mating was the second most adopted technology, although its adoption rate is considered low (29.50%), and, it also presented the highest coefficient of variation (20.85%). Other technologies such as Oestrus detection (21.41%), pregnancy diagnosis (15.40%), breeding policy (12.27%), and semen fertility evaluation (11.49%), presented very low adoption rates, being evaluation of female body condition (2.61%), the technology with the lowest adoption rate of the group.
Table 2. Reproductive technologies in dual-purpose cattle farms.
|
Code |
Technologies |
Mean |
SD |
CV |
|
T32 |
Breeding soundness evaluation in bulls |
96.61 |
18.13 |
3.29 |
|
T37 |
Type of mating |
29.50 |
45.67 |
20.85 |
|
T35 |
Oestrus detection |
21.41 |
41.07 |
16.87 |
|
T36 |
Pregnancy diagnosis |
15.40 |
36.15 |
13.07 |
|
T38 |
Breeding policy |
12.27 |
32.85 |
10.79 |
|
T33 |
Semen fertility evaluation |
11.49 |
31.93 |
10.20 |
|
T34 |
Evaluation of female body condition |
2.61 |
15.97 |
2.55 |
1Standard deviation, 2Coefficient of variation.
3.1. Social Network Analysis results
Figure 2 shows the dual-purpose farmer's technological innovation network in the reproductive area. To facilitate network visualization and interpretation, only farmers with an adoption rate higher than 40% are shown. The network structure showed that GAVATT farmers occupy the central positions who are distributed in defined subgroups with other farmers of the same organization (social homogeneity), so there is no evidence of clear leadership. Besides, the structure of the network also showed that farmers from other organizations are distributed peripherally, associated to different groups of farmers (social heterogeneity). Regarding the Producer’s associations and Non-productive organization, the farmers f_522 and f_10 stood out, with a betweenness of 19.16 and 16.64, respectively. Furthermore, the farmer f_585, who was not affiliated to any organization, was connected to the network with a betweenness of 17.97, his technological strategy was very similar to that of the farmer f_325, belonging to Cooperative, with a betweenness of 17.97.
Figure 2. One-mode network visualization of farmers with a level of technological adoption greater than 40%. Nodes’ colour: organization type.
Farmer code (f_ni), organization type- GGAVATT, SPR, Cooperative, Producers associations, No-affiliation, Non-productive organization.
Figure 3. One-mode network visualization of farmers with a level of technological adoption greater than 70%. Nodes’ colour: organization type.
Farmer code (f_ni), organization type- GGAVATT, SPR, Cooperative, Producers associations, No-affiliation, Non-productive organization.
To facilitate central farmers and technology leader’s identification, only farmers with an adoption rate higher than 70% are shown in Figure 3. In doing so, four leaders with an adoption rate of 100% were identified in the reproductive technological innovation network, all affiliated to the GGAVATT: f_1306, f_824, f_501 and f_510, with the highest betweenness (86.76) and levels of constraints above average (0.0134). The farmers' behaviour was analysed based on the established technological capacity-building, compared to other studies that have evaluated farmers' technological preferences based on adoption intentions [26,42,49,50]. Besides, f_517 was the only farmer from an organization other than GGAVATT that appears on the network. This farmer was affiliated to SPR, being the leader of this group with an adoption rate of 71.43%, and high betweenness centrality (66.69) much higher than the betweenness of other GGAVATT farmers with lower adoption rates.
Figure 4 showed the two-mode network visualization of farmers and technologies. The network structure indicated that breeding soundness evaluation in bulls (T32) is a basic technology within the reproductive technology package, adopted by most dual-purpose farmers. Another visible adoption pattern is the joint adoption of semen fertility evaluation (T33), oestrus detection (T35), and pregnancy diagnosis (T36), for which they are considered complementary technologies.
.
Figure 4. Two-mode network visualization of farmers and technologies.
Technologies- T32. Breeding soundness evaluation in bulls, T33. Semen fertility evaluation, T34. Evaluation of female body condition, T35. Oestrus detection, T36. Pregnancy diagnosis, T37. Type of mating, T38. Breeding policy; Farmers.
3.3. Benchmarking Analysis
The benchmarking of influencer farmers by organization type is shown in Table 3. The leaders were selected according to technological adoption and centrality measures (degree, betweenness, eigenvector, closeness, and constraint). The analysis showed that the farmers f_1306 and f_510 were subsistence farmers with more than 70 years and low productive indices, being the first of them meat production-oriented, and the second one milk production-oriented. The farmers f_824 and f_501 are middle-aged producers located in the wet tropic, over 45 years old, who generate two jobs. The first of them was milk production-oriented (1428,57 l/cow/year), and the second showed a mixed productive strategy towards milk and meat. These farmers were affiliated with GGAVATT and showed a high preference for reproductive technologies with an adoption rate of 100% in this technological area. Finally, farmer f_517 affiliated to the SPR was younger (31 years old), showed an adoption rate of 71.43% and a different strategy, combining the sale of cheese and meat.
Table 3. Farmer’s benchmarking by organization type and technological level.
|
Farmer, code |
f_1306 |
f_824 |
f_501 |
f_510 |
f_517 |
|
Organization type |
GGAVATT1 |
GGAVATT1 |
GGAVATT1 |
GGAVATT1 |
SPR2 |
|
Technological level, % |
100 |
100 |
100 |
100 |
71,43 |
|
Structural characterization |
|
|
|
|
|
|
Ecological zone, tropic |
Dry |
Wet |
Wet |
Wet |
Dry |
|
Productive animals, cows |
12 |
14 |
14 |
10 |
15 |
|
Animal unit, heads |
22.8 |
30.8 |
20.5 |
19.20 |
21.9 |
|
Stocking rate, UA/ha |
1.333 |
0.993 |
0.891 |
0.96 |
2.701 |
|
Grazing surface, ha |
17 |
21 |
20 |
20 |
8 |
|
Productive orientation |
Meat/ subsistence |
Milk |
Milk/ meat |
Milk subsistence |
Meat / milk |
|
Milk production, l/ha |
317.64 |
952.38 |
650 |
605 |
960 |
|
Milk yield, l/year |
5,400 |
10,000 |
6,500 |
6,050 |
7,680 |
|
Milk per cow, l/cow/year |
450 |
1,428.57 |
1,181.82 |
1,210 |
512 |
|
Calves sold, no calves |
5 |
3 |
6 |
4 |
10 |
|
Cheese yield, kg/farm/year |
0 |
0 |
0 |
0 |
1,000 |
|
Stakeholder’s age, years |
74 |
57 |
46 |
76 |
31 |
|
Dependent relatives, no |
1 |
2 |
5 |
3 |
3 |
|
Employees, workers |
0 |
2 |
2 |
2 |
1 |
1Livestock Groups for Technological Validation and Transfer; 2The Rural Production Society.
- Discussion
Dual-purpose cattle farms were very small, limited to subsistence farming where an important part of the production was oriented to self-consumption, with a highly variable productive strategy (cheese yield, milk, and calves), depending on the immediate environment and the opportunity costs. Given the context in this group of farms, the productive orientation will influence the technology adoption process [10,16,18]. Likewise, affiliation to an organization was considered an important factor in the technology adoption process [6], since organizations are places of access to technological innovations and interaction among farmers, where communication flows, constituting a determining factor in the technology capabilities of absorption, integration, and adoption.
The centrality measurements obtained were similar to that of Walther et al. [47], who used SNA to measure the effects of income and gender on informal social networks in the rice value chain, among 490 farmers in Benin, Niger and Nigeria, and found a degree of 2.5 for women and 3 for men. Our results also agree with the author in eigenvector, both for women (0.03) and for men (0.05). However, regarding betweenness, our results just match the betweenness of 92 found for women, but differ from the 170 found for men. Walter et al. [47], concluded that the monthly profit of farmers was determined by their structural position within the network and the capacity of building connections with other communities outside their ethnic groups and countries. Apart from this, Aguilar-Gallegos et al. [48], used the SNA approach to study the effects of direct and indirect interactions among 180 rubber producers and key players in the Mexican state of Oaxaca, and found a very low level of betweenness amongst farmers (0.74). This result coincides with 25% of the farmers evaluated in this research, who showed betweenness close to zero. The author relates this behaviour to the fact that some farmers establish links with extension agents rather than with their peers.
The network structure obtained provided a real diagnosis of the most vulnerable farmers technological situation, and points out the wide technological gap amongst reproduction technologies. These results were consistent with structural hole theory and network positioning studies, as high levels of constraints are associated with redundancy in contacts who provide little new information. This circumstance leads farmers to maintain their adoption practices, specializing too much in a specific technological area [2].
Two-mode network makes a significant contribution to advancing the understanding of the reproductive technologies low adoption rate as a strategic tool for development and increasing farms competitiveness and viability [9,51]. Based on the adoption patterns identified through SNA, T33, T35 and T36 are complementary technologies. It should be noted that the adoption of one of these technologies will have an impact on the adoption of the other two [6]. In this process, farmers need to know about these technologies to start implementing them on farms (knowledge and absorption capabilities) [18,19]. In this regard, the GGAVATT technicians have made a great effort to enhance the adoption of these technologies through technology transfer programs [54]. In a second stage the implementation of T34, T37 and T38 should be favoured, which requires a productive structure for their adoption [13]. These findings are aligned with those of Aguirre et al. [53], who applied SNA to study the adoption patterns of conservation agriculture practices among 222 maize smallholder farmers in the Mexican state of Chiapas, and found that farmers apply practices that solve emerging problems in the short term, but set aside those practices, which in the medium and long term, would lead to higher and stable yields. However, the network structure differs from that found by Villarroel et al. [6], who evaluated the usefulness of SNA to characterize technology leaders in small dual‐purpose cattle farms in Mexico, in the area of genetics and found a more diverse technological strategy.
The benchmarking analysis showed that farmers affiliated with GGAVATT had competitive advantages in reproduction, such as the access to knowledge (knowledge and absorption capabilities), favouring the technology adoption rate, and a high degree of connectivity with other farmers who belong to the same group [40]. The results showed a centralized pattern of technology adoption (top-down diffusion), where technological adoption, although very low, was concentrated in GGAVATT farmers, with a diffuse network structure and a differentiated leaders' network position. However, these findings differ from Villarroel et al. [6], who found a network structure with well-defined leaders who acted as knowledge disseminators in the organizations and between organizations, identifying decentralized patterns in the diffusion of technology (peer to peer). According to Dhehibi et al. [55] and Zarazúa et al. [25], being connected to technological leaders and farmers affiliated to different organizations (diverse social capital), positively affects the technology adoption rate. Besides, Zarazúa et al. [25] applied SNA to measure Social capital amongst corn producers in Mexico and found a strong relationship between the enhancement of technological innovation and the links established by farmers with actors involved in livestock activities. Farmers belonging to the most open network presented better productive outcomes and that intermediaries with greater links defined the sources and types of innovation and activate interaction patterns between several actors of the system.
Regarding the theory of dynamic capabilities, Bastanchury at al. [18], and De-Pablos-Heredero et al. [19], studied the stages of the technology adoption process in dairy sheep family farms in Spain, and described the first as a phase of technology knowledge and absorption, the second as an integration stage, and the phase of innovation capability as the final stage. The GGAVATT of a public nature and with the aim of social benefit acts as a knowledge diffusor in the first stage (knowledge and absorption capabilities), with a technology diffusion centralized model (top-down diffusion) and oriented to give technical assistance to the farmers who are affiliated to this organization [44,49]. Due to the farmer's lack of these reproductive technologies, their knowledge and absorption are carried out directly and homogeneously from the GGAVATT technicians to the affiliated farmers through participatory workshops and in situ demonstrations [54]. In the context of innovation dissemination, this is related to the social learning theory which states that people learn with and from others by example or through observation [56]. Dhehibi et al. [55] pointed out that the know-how influenced the adoption level.
In a second stage of growth and specialization, as described by Villarroel et al. [6] and Aguilar-Gallegos et al [48], early adopters present a well-defined network structure and are well-positioned within the network, standing out as leaders amongst farmers (integration capability). Probably, in a later stage of technological maturity, farmers will start a decentralized diffusion (bottom-down), resulting in innovation processes amongst farmers (peer to peer).
According to the literature and the results, the technology adoption process was sequential and cumulative. In the earlier phase, producer organizations were priorities to technology capabilities of knowledge and absorption. In the technology phase of maturity, capabilities of integration and innovation were provided by the producers, in a decentralized process with highly influential leaders. Thus, the presented findings seem to be consistent with previous research on agricultural technology adoption [6,14,16], which found that farmers’ decisions to innovate were based in the context of social interactions among themselves and with agents that promote change.
In summary, technological adoption analysis was carried out regardless of farms size, with a minimum cost strategy [5,22,51]. Due to the low level of technology adoption, the system could be improved by favouring the knowledge and absorption capabilities in the less adopted reproductive technologies [9,18]. Therefore, making it possible to increase productivity while maintaining diversified strategies (milk, meat and cheese) and land-sharing practices in DP farms.

Round 2
Reviewer 2 Report
Thank you for the opportunity to review your revised manuscript, and particular thanks for your care and attention to the comments made on the previous draft. I think your changes have made the paper clearer and also serve to increase the potential impact of your work among readers of LAND.
I only found one spelling correction. In line 235 it says "Coecient of variation", which should, of course, be "Coefficient of variation" (as it is on lines 180 and 248); this can be dealt with in copy editing and proof reading
Author Response
Suggested corrections have already been made.
I thank you for the support you have given us, which will surely improve the manuscript.
The authors